# Lensless Optical Encryption of Multilevel Digital Data Containers Using Spatially Incoherent Illumination

**Pavel Cheremkhin, Nikolay Evtikhiev, Vitaly Krasnov \*, Ilya Ryabcev, Anna Shifrina and Rostislav Starikov**

Laser Physics Department, Institute for Laser and Plasma Technologies, National Research Nuclear University MEPhI (Moscow Engineering Physics Institute), Kashirskoe Shosse 31, 115409 Moscow, Russia; cheremhinpavel@mail.ru (P.C.); k1121@mail.ru (N.E.); ilja.rjabcev0@gmail.com (I.R.); avshifrina@gmail.com (A.S.); rstarikov@mail.ru (R.S.)
\* Correspondence: vitaly.krasnov@mail.ru

**Abstract:** The necessity of the correction of errors emerging during the optical encryption process led to the extensive use of data containers such as QR codes. However, due to specifics of optical encryption, QR codes are not very well suited for the task, which results in low error correction capabilities in optical experiments mainly due to easily breakable QR code's service elements and byte data structure. In this paper, we present optical implementation of information optical encryption system utilizing new multilevel customizable digital data containers with high data density. The results of optical experiments demonstrate efficient error correction capabilities of the new data container.

**Keywords:** optical encryption; digital data containers; spatially incoherent illumination; spatial light modulator; diffraction optical element; divergent beams; QR code





## 1. Introduction

Optical encryption is a popular area of scientific research based on the double random-phase encoding technique (DRPE), created by Refregier and Javidi in 1995 [1]. Over the next 25 years, new optical encryption methods were developed using the same principles [2–4]. These techniques can utilize different transforms of optic fields (fractional Fourier [5–10], Fresnel [11–14], Hartley [15–18], etc.), use different kinds of phase masks [19–24] or metasurfaces [25–27], be asymmetrical [28–32] or be used for authentication and image hiding [33–40].

In DRPE, the ciphertext is a complex amplitude; therefore, for lossless registration, it requires digital holographic techniques and a fully coherent illumination. As a result, a speckle noise occurrence in the decoded image is inevitable. To eliminate speckle noise pollution Barrera et al. proposed in [41] to use a data "container" in the form of quick response (QR) codes. This idea quickly became popular in the optical encryption studies [42–45]. However, QR codes were originally developed for machine vision systems, not for cryptosystems, and are not optimal for them—the main issues being: presence of the positioning elements, use of byte code where minimal code unit can be presented by 8 pixels at least, small maximum size (177 × 177), and only three available error correction levels. Using similar to QR codes ideas, some attempts to create new data containers have been made [46,47].

Earlier, we proposed the new customizable digital data container (CDDC) based on the use of Bose–Chaudhuri–Hocquenghem error-correction codes (BCH) [48]. CDDC can streamline the practical implementations of an optical encryption by allowing one to use modern optical hardware to its full capabilities by fine-tuning the parameters of CDDC. One of the important parameters is a number of gray levels in a container. For example, an increase from two gray levels to four results in two times increase in data density. Using a multilevel data container instead of binary one (such as QR code), we can significantly improve encryption system's bandwidth. However, multilevel containers are more fragile

and sensitive to noise pollution. To analyze and calculate multilevel customizable digital data container (MCDDC) error correction capabilities, we conducted a series of numerical and optical experiments; the results of which are presented in this paper.

The rest of the paper is organized as follows: Section 2 discusses data containers for an optical encryption, such as QR codes and MCDDC. Section 3 provides a detailed description of the algorithm of data packing and unpacking for MCDDC. Section 4 presents the results of numerical simulation of MCDDC application in an optical encryption system. In Section 5, the use of MCDDC in an experimentally implemented optical encryption system with spatially incoherent illumination is demonstrated.

## 2. Data Containers for an Optical Encryption

### 2.1. QR Code as a Data Container

To the best of our knowledge, the only full-fledged data container, commonly used in an optical encryption, is QR code. QR codes were developed as 2D bar codes for machine vision systems (as pointed in [3]). The design of QR codes was determined by the requirements of their application: they have to have high resistance to damage (up to a partial loss) and should be easy to read in various conditions.

The robustness of QR codes against damage is provided by the use of the Reed–Solomon error correction codes (RS). In the same way as BCH, RS can maintain the integrity of the data within and correct errors that occur. This ability is realized by adding extra information to the block of the useful data. The number of correctable errors is determined by the parameters of a code (the length of the input data block and the volume of extra information).

An important difference between RS and BCH is their format: RS codes operate in the byte format, while BCH—in the bit one. The byte format is a major disadvantage of QR codes, when they are used in an optical encryption. For binary QR code, a byte format of RS means that it needs 8 modules (QR code elements) to represent minimal fragment of encoded data visually. If even one module within is damaged, the whole 8-modules fragment is incorrect [46].

QR code has only four available error correction levels of approximately 7%, 15%, 25% or 30%; however, due to the byte format, these percentages represent not errors in modules but in whole blocks of eight of them, resulting in much lower error correction capabilities than expected [48].

Other shortcomings of QR codes include an abundance of service elements, associated with positioning and orientation. Such elements are used for the correction of QR code placement before its decoding and are unnecessary for an optical encryption because it rarely leads to notable image shift or rotation. These elements reduce the area available for data recording and do not allow the decoding of QR code in case of their damage, although the error correction limit may not yet be reached.

Secondly, common QR codes have a small range of available-to-user parameters. Maximum size is $177 \times 177$ modules with strictly 1:1 aspect ratio. The error correction levels are limited to four fixed values. The maximum number of interconnected codes, which contain one file, is 16; therefore, their maximum data volume is only about 50 kB.

### 2.2. MCDDC's Features and Parameters

We developed MCDDC using the same principles as QR codes but removed and changed all undesirable elements. The most important difference is a use of BCH instead of RS. BCH operates in binary format, and its minimal fragment is strictly one module. Therefore, one incorrect module in BCH corresponds to one lost bit, instead of eight in RS.

Since modern data content is measured in gigabytes, MCDDC by default is a sequence of messages. We discarded all patterns associated with positioning and orientation and combined all service elements in one block to maximize useful data area. The service block was put into the center of the first message, with the following messages containing only encoded data. The structure of MCDDC's messages is presented in Figure 1.

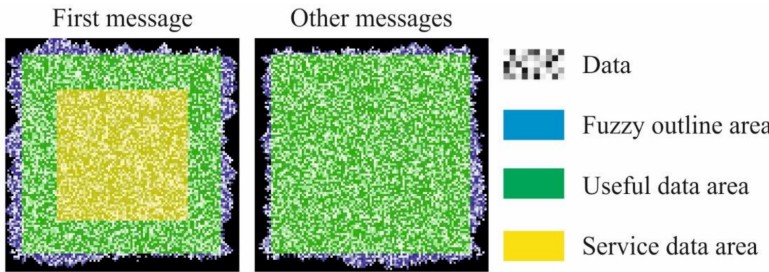

**Figure 1.** The structure of MCDDC's messages (color online).

In general, all MCDDC messages have a fuzzy outline instead of straight borders. A fuzzy outline is an advantage for data container, when it is used in optical encryption systems that do not transform an input image spectrum into a white noise. In these systems, it is possible to trace outlines of an input image in an encoded one; therefore, straight outlines can give out the size of the input image.

The service block has a size of $52 \times 64$ modules and contains all the necessary information about parameters of MCDDC. There are 10 sub-blocks, one for each parameter; the most important of which are:

- Full size of a message (taking into account the width of the fuzzy outline if it is present).
- BCH error correction level. This parameter shows the error correction capabilities of the used code, not MCDDC itself. MCDDC error correction level is lower due to the normal distribution of the noise in a message and can vary for different encryption systems.
- Number of gray levels in messages.
- Width of the fuzzy outline or its absence.
- Number of interconnected messages.

Another important difference between MCDDC and QR code is a data scrambling. RS and BCH encode data by blocks with typical size of tens and hundreds of bits. In MCDDC, after all non-service data are encoded, they are scrambled. It allows to mitigate the effect of errors localization. If there is a cluster of errors, scrambling spreads them over a number of blocks. It drastically reduces the likelihood that a number of errors in any one block prematurely exceed the BCH error correction capabilities.

The full range of available-to-user MCDDC parameters and its comparison to QR code are presented in Table 1.

**Table 1.** Comparison of QR code and MCDDC features.

| Feature | QR Code | MCDDC |
|---|---|---|
| Available sizes | 40 values in the range 21–177 modules | All values in the range 64–8192 modules |
| Aspect ratio | 1:1 | Any value |
| Input error correction level | 7, 15, 25 or 30% | 1–21.7% |
| Container experimental error correction level | 0.7% | 0.5–16.3% |
| Density of data encoding | 0.33–0.75 | 0.08–0.86 |
| Maximum number of connected messages | 16 | 8192 |
| Data scrambling | No | Yes |
| Large structured blocks (positioning and orientation patterns) | Present | Absent |
| Number of gray levels in messages | 2 | Any value |
| Outlines | Straight | Fuzzy |

To determine container experimental error correction level, we used noise pollution simulation with "salt and pepper" kind of noise and "one module—one pixel" container representation when single QR code module corresponds to a single pixel in an image [48]. The low results of QR code (the only one available value of 0.7%) are due to its unsuitability for such representation. In experiments with coherent illumination, one module of QR code is usually represented by a block of several dozens of pixels, but it led to a very low density of data encoding. Due to the bit format of BCH, MCDDC can successfully operate in this representation.

Due to the availability of high error correction levels, it is possible to use MCDDC with multiple gray levels. User can choose a number of gray levels in accordance with their experimental setup's signal-to-noise ratio (SNR). The increasing of a number of gray levels correspondingly increases a density of data encoding. As an example, in Figure 2 a file (format txt, size 3 kB) is presented, encoded in a sequence of five binary QR codes or two CDDC grayscale messages with four gray levels; useful data area for QR code and CDDC were chosen as the same, 97 × 97 modules.

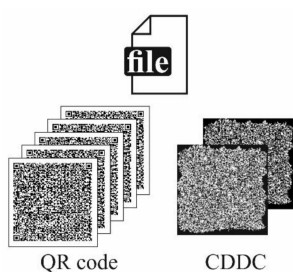

QR code CDDC

**Figure 2.** Higher density of data encoding in MCDDC compared to QR code.

However, as was mentioned above, grayscale containers are more fragile. If each module of MCDDC contains several bits of data, one error leads not to one lost bit but to several of them. It is important to find a balance between the higher density of data encoding and the increased requirement for the system's signal-to-noise ratio.

### 3. Algorithms of Data Packing and Unpacking for MCDDC

The MCDDC data packaging process is the first step of data encryption process. Input digital information is packed into a series of MCDDC, which are then displayed on the first LC SLM. The process is illustrated by Figure 3.

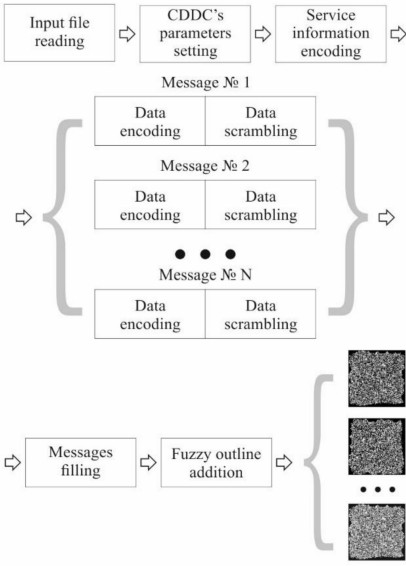

**Figure 3.** The algorithm of MCDDC data packing.

Step 1. Parameters Setting and Definition

At this step, the user sets the parameters of MCDDC: the message size, BCH error correction level, the number of gray levels, and the width of the fuzzy outline and also selects the file with data for packing. According to the selected values, BCH coefficients are determined. After reading the file, its length in bits is determined; based on which and taking into account the specified parameters of MCDDC, the number of messages is calculated.

Step 2. Data encoding and message filling

All the information received at the previous step, both service and useful data, is converted into a bit format.

The bit values of the parameters are encoded by highly error-tolerant BCH {255,13} (up to 59 errors out of 255 bits can be corrected in any single block). Then, the encoded sequences are assembled into a $52 \times 64$ block, which is placed in the center of the first message.

The data for packaging is divided into fragments, the length of which is determined by selected BCH, and is encoded.

After all the data have been encoded, they are combined into a single string, translated into a multibit representation (correspondingly to the selected number of gray levels) and scrambled. Messages are filled in line by line, starting from the upper-left corner.

Step 3. Applying a fuzzy outline

A fuzzy outline with the number of gray levels corresponding to the data area is added to each message (if the use of a fuzzy outline is selected). The outline is generated randomly; its maximum width corresponds to the value selected by the user.

The unpacking is the last step of data decryption process. It is performed after an encrypted MCDDC image is decrypted, its raster transformed to the original one, and quantization is performed. It is a reverse to the packing process which allows one to extract digital information from the decrypted image. The process is illustrated by Figure 4.

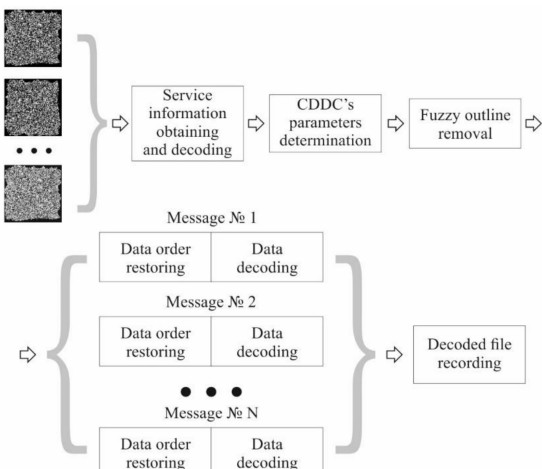

**Figure 4.** The algorithm of MCDDC data unpacking.

Step 1. Obtaining encoded service information from the first message

Since the service information block is located in the center of the first message and has a fixed size, the information can be extracted even without knowing the exact size of the message or a fuzzy outline width.

Step 2. Decoding of the service information and determination of MCDDC parameters

The obtained information is decoded and checked to see if all the errors that occurred during decoding were corrected (BCH allows to gather information about the presence of uncompensated errors). In the case of error-free decoding, MCDDC parameters are determined.

Step 3. Data decoding

Based on the parameters values, obtained during Step 2 (message size), a fuzzy outline is discarded. Encoded useful data are read from the messages. The correct order of the data is restored. For multilevel grayscale CDDC, a return to the binary representation is performed. Next, the data are divided into fragments in accordance with BCH parameters and decoded.

Step 4. Recording of the reconstructed file

After decoding all messages, the data extracted from them are collected into a single file. The resulting file is recorded.

Unlike for the service block, the occurrence of uncorrected errors in the useful data does not interrupt the execution of the algorithm. If the number of such errors is small, then the packed file can be reconstructed, although with some damage.

## 4. Noise Robustness Analysis

For practical implementations, the most important characteristic of MCDDC is its error correction level and density of data encoding.

Using noise pollution simulation for testing purposes, we determined how the container experimental error correction level depends on the normalized standard deviation (NSTD) value for MCDDC with a different number of gray levels. We used Gaussian noise with gradually increasing variance and "one module—one pixel" representation to determine the noise level before the first uncorrected errors occurred. This threshold value is the container experimental error correction level.

For numerical simulation, we used $3 \times 30$ sets of MCDDCs (with two, four and eight gray levels). Each MCDDC contained a text file in txt format of 5–30 kB size (the size was chosen so that after file encoding, no more than 10 MCDDC messages were received). MCDDC parameters were: message size—$255 \times 255$ modules, fuzzy outline width 10% and BCH input error correction level in the range 1–21.7%. For statistics, five noise distributions were generated for each MCDDC.

The results of the simulation are presented in Figures 5 and 6. For comparison, we presented results for QR code with similar parameters (black circle).

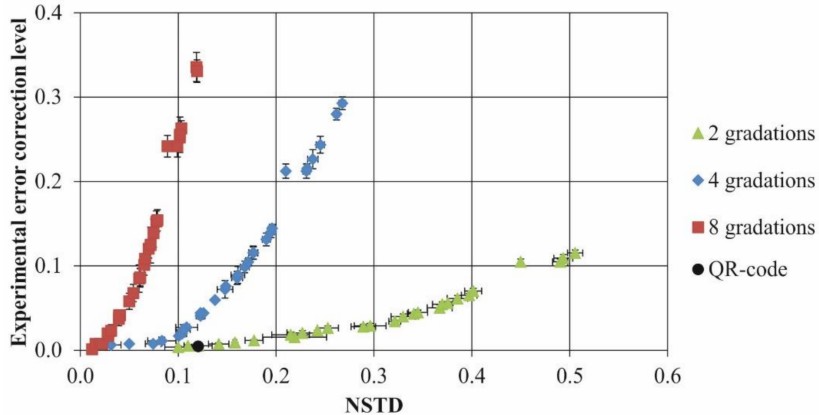

**Figure 5.** Dependencies of the experimental error correction level on the NSTD value for MCDDC with different number of gray levels (color online).

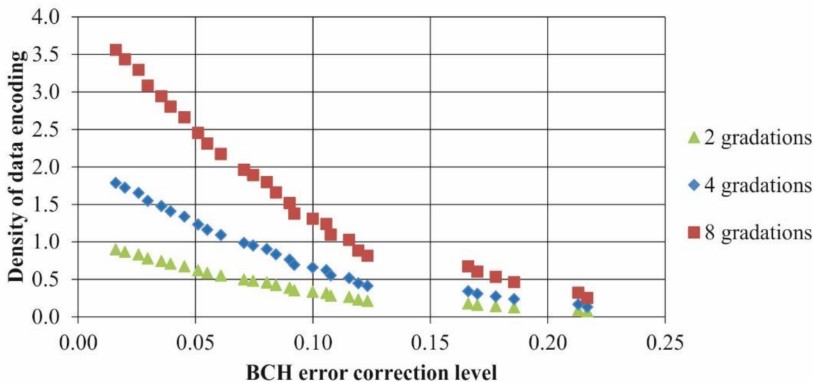

**Figure 6.** Dependencies of the density of data encoding on BCH error correction level for MCDDC with different number of gray levels (color online).

As can be seen, experimental error correction levels lie in vastly different ranges for MCDDC with different number of gray levels. Binary MCDDC can withstand the maximal level of noise pollution but has a lower level of density of data encoding.

It is important to note that MCDDC can have a range of error correction levels bigger than for input error correction levels. This is caused by the fact that noise errors usually lead to a deviation of a gray level per module by 1. However, each module contains several bits of information (in this case, 2 or 3 bits), and only one of them is damaged; the rest are "protected" from damage.

## 5. Optical Experiment

For experimental testing of MCDDC, the lensless optical encryption scheme with spatially incoherent illumination based on two LCoS spatial light modulators (SLM) was used (see Figure 7). For description of packing/unpacking procedures, see Section 3.

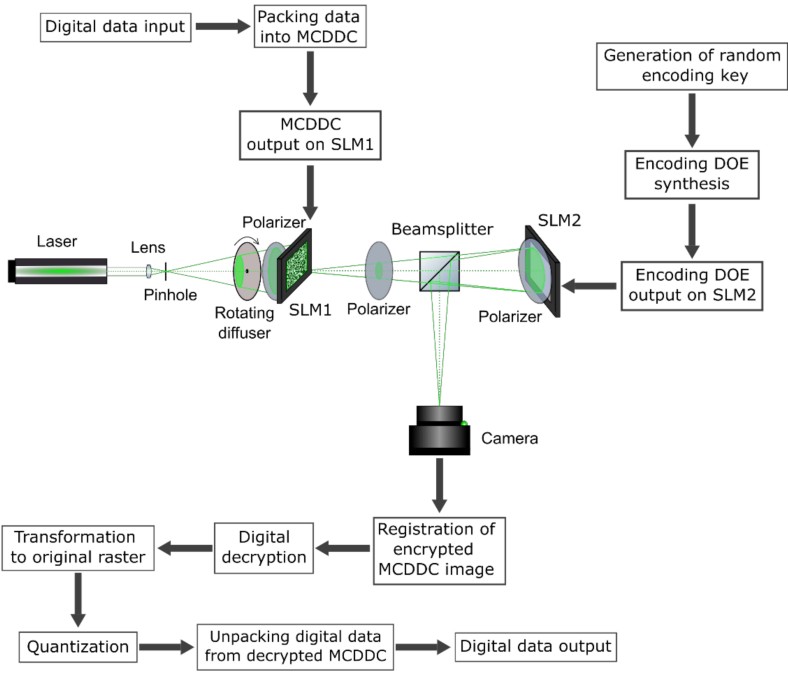

**Figure 7.** The experimental scheme of MCDDC lensless optical encryption with spatially incoherent illumination.

Nd-YAG laser with a wavelength of 532 nm was used as a light source. The filtered divergent light beam is formed by a combination of a lens and a pinhole. The pinhole was

placed in the focus of microscope lens in order to filter possible low-frequency wavefront distortions which can lead to inhomogeneity of input plane illumination. The rotating diffuser destroys spatial coherence of radiation and suppresses speckle-noise due to the formation of a new speckle pattern every time its rotation results in new grain structure been illuminated. Its rotation speed should be high enough so that during frame exposure time, many speckle patterns would be averaged, effectively suppressing speckle noise. In our setup, the diffuser rotation speed was equal to 1200 rpm, its radius—5 cm and typical grain size—10 μm. The amplitude LCoS SLM HoloEye LC2012 with $1024 \times 768$ pixels (SLM1; pixel size is $36 \times 36$ μm) was used for the display of MCDDC. The phase LCoS SLM HoloEye GAEA-2 with $4160 \times 2464$ pixels (SLM2; pixel size is $3.74 \times 3.74$ μm) was used for the display of encoding diffraction optical elements (DOE). The beamsplitter cube reflects light from the SLM2 to the Flare 48MP camera with a resolution of $7920 \times 6004$ pixels (pixel size is $4.6 \times 4.6$ μm) which was used for the registration of encrypted images. Polarizers were used for the elimination of depolarization introduced by the diffuser, SLM1 and SLM2. Exposure time was set to 33 ms, and images were registered with a maximum signal close to saturation, but not reaching it. The necessary level of illumination was acquired by regulating the laser power.

The DOE's focal lens and distances determine system's optical magnification according to thin-lens formula, as well as the size of encoding DOE's point spread function and, therefore, the size of an encrypted image. In addition, the optical distance between SLM2 and camera must be known in order to construct the decoding filter as described in [49]. The tolerance for this distance when constructing decoding filter, according to our experiments on similar setup in [49], is ±0.5%. The optical distance between SLM1 and SLM2 was 34 cm, between SLM2 and the camera—38 cm, and DOE's focal length—19 cm. An image of a single SLM1 pixel occupies a section of $10.1 \times 10.1$ pixels of the camera. One element of encoding DOE PSF occupies a section of $2.6 \times 3.7$ pixels on the camera.

MCDDCs with $128 \times 128$ elements and $254 \times 254$ elements with four gray levels and with the input error correction level of 21.7% were used in the experiments (see Figure 8). The size of the fuzzy outline area for containers with $128 \times 128$ elements is 10% of the linear size of the container on each side, and for containers with $254 \times 254$ elements—25% of the linear size of the container on each side. Encryption was carried out with two encryption keys with $128 \times 128$ elements with densities that were characterized by normalized average energy (NAE) 0.0005 and 0.001 and with two gray levels. The encryption keys with different NAE values and corresponding encoding DOEs are shown in Figure 9. The size of the encoding DOEs was $4000 \times 2464$ elements.

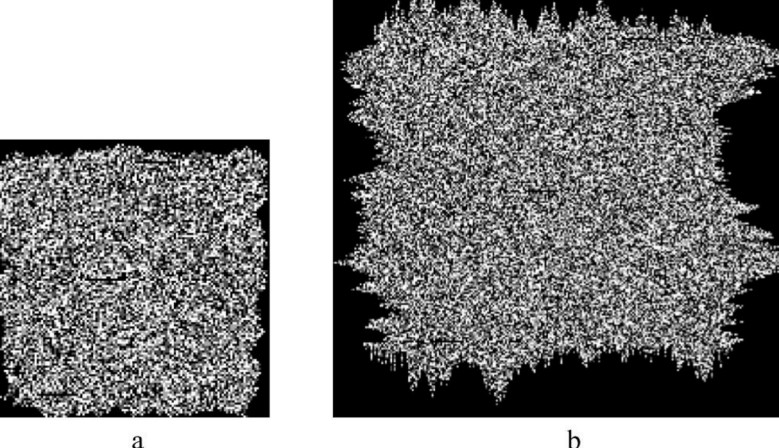

**Figure 8.** The test MCDDC images with $128 \times 128$ elements (**a**) and $254 \times 254$ elements (**b**).

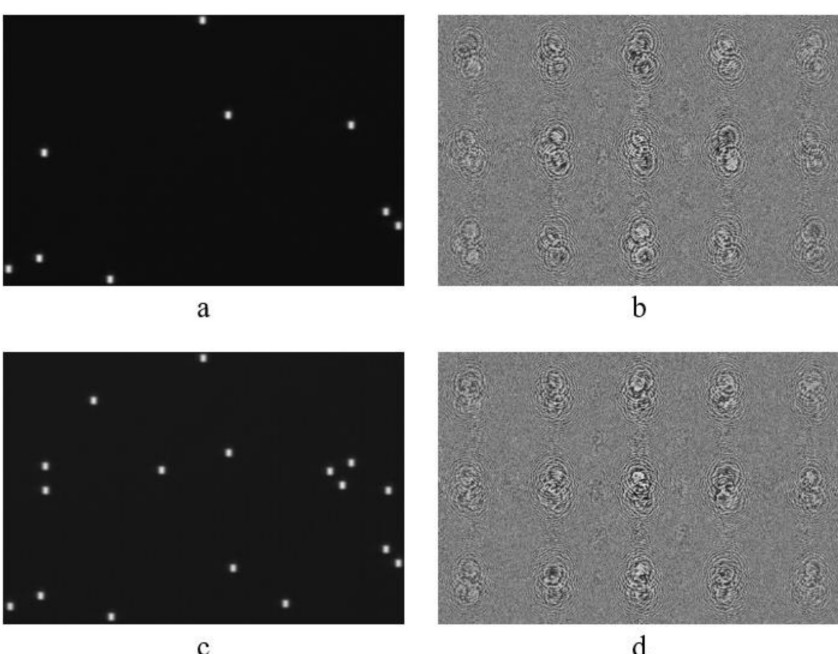

**Figure 9.** The encryption keys with NAE 0.0005 (**a**) and 0.001 (**c**) and corresponding DOEs (**b**,**d**).

The number of incorrect pixels in the decrypted image was used as a measure of decryption error. The original images, encrypted images and keys were averaged over sixteen images to suppress noise. The images encrypted using the key with NAE 0.0005, and corresponding decrypted images are shown in Figure 10.

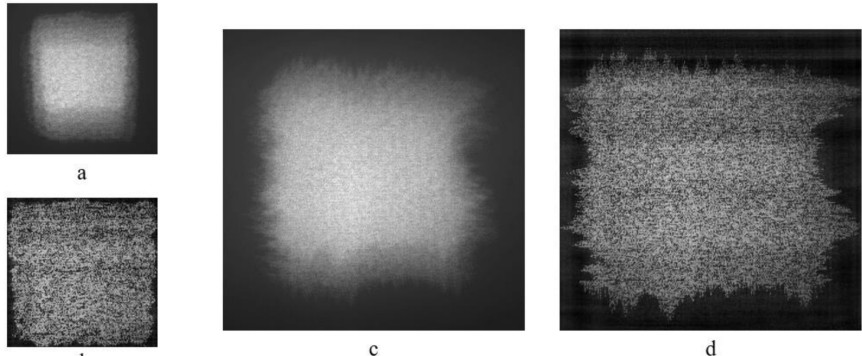

**Figure 10.** Encrypted (**a**,**c**) and decrypted images (**b**,**d**) corresponding to MCDDC with 128 × 128 (**a**,**b**) and 254 × 254 (**c**,**d**) elements.

Decryption was carried out by inverse filtering with Tikhonov regularization [50]. After that, the decrypted images were quantized. Due to the nonuniformity of image histogram, nonuniform quantization was used [51]. Image histogram was divided into zones with equal pixel quantity. The quantity of quantization levels is the number of these zones. The quantized decrypted images and error distribution maps are shown in Figure 11. The dark points in error distribution map correspond to the pixels with gray level lower than correct ones, bright point—pixels with gray level above correct one.

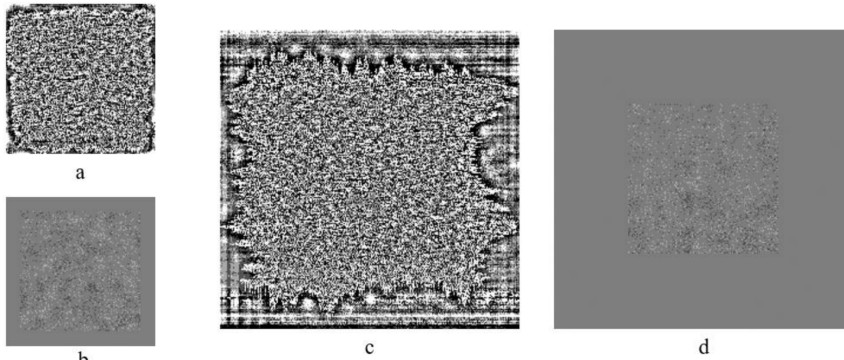

**Figure 11.** Quantized decrypted images (**a**,**c**) and error maps (**b**,**d**) for MCDDCs with 128×128 (**a**,**b**) and 254 × 254 (**c**,**d**) elements.

The numbers of incorrectly decrypted pixels and the error rates for various keys and MCDDC sizes are presented in Table 2.

**Table 2.** Numbers of incorrectly decrypted pixels and error rates.

| Key NAE | MCDDC Size | |
| --- | --- | --- |
| | **128 × 128** | **254 × 254** |
| 0.0005 | 1925 (18.4%) | 2392 (14.8%) |
| 0.001 | 1944 (18.5%) | 2230 (13.8%) |

In the experiments, the information was successfully extracted from all decoded MCDDCs presented in Table 2. The minimum achieved error rate was 13.8%. It is necessary to set the input error correction level at least 1.5 times greater than the actual error rate to avoid the probability of data decoding failure. In our case, it is appropriate to use the error correction level of 20%, which corresponds to the density of data encoding of 0.24.

## 6. Security Analysis

In order to assess the security of the implemented setup, we had performed the analysis of sensitivity of encryption key to distortions. For the analysis, we used MCDDC with 128 × 128 elements (see Figure 8a) encrypted (see Figure 10a) using the key with NAE 0.001, which contains 19 nonzero elements (see Figure 9a).

We relocated single nonzero element and made an attempt at decrypting an encrypted image from Figure 10a using the distorted key. Decryption resulted in 2501 errors (24% errors); decrypted MCDDC was decoded; and only 1 (0.6%) symbol out of 170 was decoded incorrectly. The relocation of two nonzero elements resulted in 3024 errors (29% errors) and led to 18 (11%) incorrectly decoded symbols. The relocation of three nonzero elements resulted in 3548 errors (34% errors) and 43 (25%) incorrectly decoded symbols and made encoded text unreadable.

As described in [49], if one possesses the information about the exact number of nonzero elements $k$, then the number of possible key combinations is:

$$C = \frac{(N_x N_y)!}{k!(N_x N_y - k)!} \tag{1}$$

For a key with 128 × 128 elements, 19 out of which are not zero, this number is equal to $9.6 \times 10^{62}$. It corresponds to a one-dimensional binary key with a 209 bit length.

Next, let us evaluate the number of acceptable decryption keys [52] which allow one to successfully decode encrypted data. In the case of MCDDC, which can partially decode text even when the damage threshold is exceeded, we consider decoding successful when the decoded text can still be read. If $r$ nonzero elements out of total $k$ can be randomly

relocated and the resulting key allows for the successful decryption of an encrypted data, then the number of acceptable decryption keys can be found as the number of possible variants of choosing of any of *(k−r)* correct nonzero elements multiplied by the number of possible variants of positioning of "unnecessary" *r* nonzero elements:

$$S = \frac{k!(N_x N_y)!}{(k-r)!(r!)^2(N_x N_y - r)!} \tag{2}$$

For our case of a key with $128 \times 128$ elements, 19 out of which are nonzero and 2 can be relocated, this corresponds to $2.3 \times 10^{10}$ combinations or 34 bit length of one-dimensional binary key.

To find one acceptable key, the number of combinations that should be tried can be found as:

$$K = \frac{C}{2S} = \frac{(k-r)!(r!)^2(N_x N_y - r)!}{2 \cdot (k!)^2 \ (N_x N_y - k)!} \tag{3}$$

which in our case makes $4.2 \times 10^{52}$ combinations. This corresponds to a one-dimensional binary key with a 175 bit length. It is a sufficiently large key to effectively withstand brute-force attacks (the minimum key length for that is 100 bit [4]).

## 7. Conclusions

In this paper, we presented new multilevel customizable digital data containers (MCDDC) with high data density designed to outperform widely used QR codes in information optical systems, such as optical encryption systems. The results of numerical experiments on noise robustness analysis demonstrate the efficiency and superiority of new MCDDC over QR codes. The results of the application of four gray levels MCDDC in optical encryption system demonstrate the efficiency of the proposed containers as all the data were decoded without a single error. In addition, the use of several gray levels instead of only two increases the sensitivity of the data containers to tampering, which increases security. Thus, we have shown the superiority of new MCDDC over binary data containers, such as QR codes, in terms of data density, flexibility and error correction capabilities.

**Author Contributions:** Conceptualization, V.K. and A.S.; methodology, V.K. and R.S.; software, V.K. and A.S.; validation, P.C. and I.R.; formal analysis, R.S.; investigation, V.K. and I.R.; resources, N.E.; data curation, V.K. and A.S.; writing—original draft preparation, V.K., I.R. and A.S.; writing—review and editing, P.C., V.K. and A.S.; visualization, V.K. and A.S.; supervision, R.S.; project administration, N.E.; funding acquisition, V.K. All authors have read and agreed to the published version of the manuscript.

**Funding:** This research was funded by the Russian Science Foundation (RSF), grant number 21-79-00117.

**Institutional Review Board Statement:** Not applicable.

**Informed Consent Statement:** Not applicable.

**Data Availability Statement:** Data underlying the results presented in this paper are not publicly available at this time but may be obtained from the authors upon reasonable request.

**Conflicts of Interest:** The authors declare no conflict of interest.

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
