# Peer review of "Lensless Optical Encryption of Multilevel Digital Data Containers Using Spatially Incoherent Illumination"

_applsci, doi:10.3390/app12010406_

Round 1
Reviewer 1 Report
In this paper the authors presented the multilevel customizable digital data containers with high data density for information optical systems, such as optical encryption systems.. The topic is interesting and matches well for MDPI Applied Sciences journal. The paper contains meaningful review of related works. However the paper has some unclear points, and the following minor concerns.
1. In my opinion, the paper would benefit significantly if the authors would semantically linked the description of packing and unpacking algorithms in Figures 3 and 4 with the optical scheme of the experiment in Figure 7. For example, they would indicate how the encoding DOE is used in these algorithms.
2. One of the goals of the work is to reduce the influence of speckle and use, in this regard, spatially incoherent illumination. Therefore, it would be useful to demonstrate the need for a rotating diffuser. And also give the results with a fixed diffuser.
3. The authors generate spatially incoherent illumination using a rotating diffuser. It can probably be assumed that the prototype is supposed to use a light-emitting diode as a light source, which forms partially coherent illumination. In this regard, it would be useful to consider how the speed of rotation of the diffuser affects the result. And how does this compare with the coherence characteristics of typical LEDs.
4. DOE abbreviation must be deciphered in the text of the paper.
Reviewer 2 Report
The extension of optical encryption of information from 1D bar code systems to 2D QR code systems took place several years ago and lead to a widespread adoption in machine vision. This paper replies to an implicit question: Could a better system than QR be envisaged?
The authors start from a thorough analysis of the basic features of a QR system and put in evidence their limitations in terms of obtainable information density and error correction capability. They then show that a new type of data container, bit instead of byte oriented, and carefully designed to remove negative features of QR systems can outperform QR in terms of reliability, information capacity, and error correction level, to quote a few. After a detailed exposition of the structure of messages adopted for the new container, its performances are assessed through numerical simulations. The method is tested with an optical setup, in which spatial light modulators play a key role. The results show explicitly the capabilities of the method.
The paper accounts for a very good piece of work and deserves acceptation with few revisions.
For a paper entitled “Lensless optical encryption…” the optical system description is rather incomplete and should be improved:
1] The pinhole put after the focusing lens has generally the role of cleaning the laser beam from artifacts coming from spurious reflections and the like. In the present case it does not have any role because the purity of the beam is anyway destroyed by the rotating diffuser.
2] Three polarizers appear in the scheme, but their role is not explained. I guess the authors write polarizer to mean linear p. Please specify. Further: which depolarizing element is present in the setup that requires their use?
3] Distances of spatial light modulators from other elements are explicitly given, but
there is no explanation of how they were chosen and of the pertinent tolerances. Please explain.
References
The suggested reference for ill-posed problems is Arsenin-Tikhonov (Ref. 49) in Russian. I’m afraid the majority of Appl. Sci. readers does not speak Russian. I remember that there existed a French translation of the celebrated text of Arsenin and Tikhonov (MIR edition). An English translation could also exist, but I’m not aware of it. If the authors are aware of an available translation in French or English they should quote it. Otherwise could the authors suggest an alternative source?
